# Task Regularized Hybrid Knowledge Distillation For Continual Object Detection

## Abstract

Knowledge distillation has been used to overcome catastrophic forgetting in Continual Object Detection(COD) task. Previous works mainly focus on combining different distillation methods, including feature, classification, location and relation, into a mixed scheme to solve this problem. In this paper, we propose a task regularized hybrid knowledge distillation method for COD task. First, we propose an image-level hybrid knowledge representation by combining instance-level hard and soft knowledge to use teacher knowledge critically. Second, we propose a task-based regularization distillation loss by taking account of loss and category differences to make continual learning more balance between old and new tasks. We find that, under appropriate knowledge selection and transfer strategies, using only classification distillation can also relieve knowledge forgetting effectively. Extensive experiments conducted on MS COCO2017 demonstrate that our method achieves state-of-the-art results under various scenarios. We get an absolute improvement of 27.98 at $RelGap$ under the most difficult five-task scenario. Code is in attachment and will be available on github.

## 1 Introduction

The existing object detection models (Ge et al., 2021) mainly adopt overall learning paradigm, in which the annotations of all categories must be available before learning. It assumes that data distribution is fixed or stationary (Yuan et al., 2021), while data in real-world comes dynamically with a non-stationary distribution. When model learns from incoming data continually, new knowledge interferes with the old one, leading to catastrophic forgetting (McCloskey & Cohen, 1989; Goodfellow et al., 2014). To solve this problem, continual learning is proposed in recent years and has made progresses in image classification (Zeng et al., 2019; Qu et al., 2021). On the other hand, **c**ontinual **o**bject **d**etection (COD) is rarely studied.

Knowledge distillation (Hinton et al., 2015) has been proved to be an effective method for COD task, in which the model trained on old classes performs as a teacher to guide the training of student model on new classes. There are four kinds of distillation schemes: feature, classification, location and relation distillation. Most previous works combine feature distillation and classification distillation to construct their distillation methods (Li & Hoiem, 2018; Li et al., 2019; Yang et al., 2022b), while the latest work (Feng et al., 2022) combines classification distillation and location distillation to construct a response-based distillation method. In addition, various distillation losses, based on KL diversity, cross entropy and mean square error, are proposed for knowledge transfer. In summary, the keys of knowledge distillation are what knowledge should be selected from teacher and how it is transferred to student. The former question needs **K**nowledge **S**election **S**trategy (KSS), while the latter needs **K**nowledge **T**ransfer **S**trategy (KTS).

Continual object detection face two problems. **(1)** Teacher outputs probability distributions as logits and converts them into one-hot labels as final predictions. Logits and one-hot labels are regarded as soft and hard knowledge, respectively. Soft knowledge contains confidence relations among categories, but brings knowledge fuzziness inevitably. While, hard knowledge has completely opposite effects. Therefore, how to design KSS to keep balance between accuracy and ambiguity of knowledge is a key problem. **(2)** Continual learning should maintain old knowledge during the learning of new knowledge to overcome catastrophic forgetting, therefore how to design KTS to keep balance between stability of old knowledge and plasticity of new knowledge is a key problem. This paper

focuses on how to design effective KSS and KTS for COD task. We demonstrate that as long as KSS and KTS are good enough, using only classification distillation can also significantly alleviate catastrophic forgetting to improve performance.

Firstly, the max confidence value of logits is always lower than its corresponding one-hot value (equal to 1), which brings knowledge ambiguity and reduces supervise ability of teacher. This means soft knowledge is not completely reliable, which should be used critically. However, previous methods ignore this keypoint. Motivated by this insight, we propose an image-level **h**ybrid **k**nowledge **r**epresentation method, named as **HKR**, by combining instance-level soft and hard knowledge adaptively to improve the exploration of teacher knowledge. Secondly, new coming data contains massive labeled objects of new classes, while contains a few unlabeled objects of old classes, therefore student trends to be dominated by new classes and falls into catastrophic knowledge. Thus it is very important to balance the learning of old and new classes. We propose a **t**ask **r**egularized **d**istillation method, named as **TRD**, by using losses difference between old and new classes to prevent student from task over-fitting effectively. We first explore imbalance learning problem explicitly for COD.

Our contributions can be summarized as follows: **(1)** We propose a hybrid knowledge representation strategy by combing logits and one-hot predictions to make a better trade-off and selection between soft knowledge and hard knowledge. **(2)** We propose a task regularized distillation method as an effective knowledge transfer strategy to overcome the imbalance learning between old and new tasks, which relieves catastrophic forgetting significantly. **(3)** We demonstrate that, compared with the composite distillation schemes, using only classification distillation with appropriate knowledge selection and transfer strategies can also reach up to the state-of-the-art performance of COD task.

## 2 RELATED WORKS

**Continual Object Detection.** There are several schemes for COD task. Li & Hoiem (2018) first proposed a knowledge distillation scheme by applying LWF to Fast RCNN (Girshick, 2015). Zheng & Chen (2021) proposed a contrast learning scheme to strike a balance between old and new knowledge. Joseph et al. (2021b) proposed a meta-learning scheme to share optimal information across continual tasks. Joseph et al. (2021a) introduced the concept of Open World Object Detection, which integrates continual learning and open-set learning simultaneously. In addition, Li et al. (2021) first studied few-shot COD. Li et al. (2019) designed a COD system on edge devices. Wang et al. (2021) presented an online continual object detection dataset. Recently, Wang et al. (2022) proposed a data compression strategy to improve sample replay scheme of COD. Yang et al. (2022a) proposed a prototypical correlation guiding mechanism to overcome knowledge forgetting. Cermelli et al. (2022) proposed to model the missing annotations to improve COD performance.

**Knowledge Distillation for Continual Object Detection.** Knowledge distillation (Hinton et al., 2015) is an effective way to transfer knowledge between models with KL diversity, cross entropy or mean square error as the distillation loss. There are mainly four kinds of knowledge distillation used in COD task: feature, classification, location and relation distillation. LwF was the first to apply knowledge distillation to Fast RCNN detector (Li & Hoiem, 2018). RILOD designed feature, classification and location distillation for RetinaNet detector on edge devices (Li et al., 2019). SID combined feature and relation distillation for anchor-free detectors (Peng et al., 2021). Yang et al. (2022b) proposed a feature and classification distillation by treating channel and spatial feature differently. ERD is the latest state-of-the-art method, combining classification and location distillation (Feng et al., 2022). Most of existing methods combine feature, classification and location distillation in composite and complex schemes to realize knowledge selection and transfer.

## 3 OUR METHOD

### 3.1 OVERALL ARCHITECTURE

We build our continual object detector on the top of YOLOX (Ge et al., 2021). Fig1 shows its overall architecture. YOLOX designs two independent branches as its classification and location heads. Firstly, hybrid knowledge selection (**HKS**) module works after the classification head of teacher to discover and select the valuable predictions for old classes. Secondly, task regularized distillation (**TRD**) module works between the heads of teacher and student to transfer knowledge.

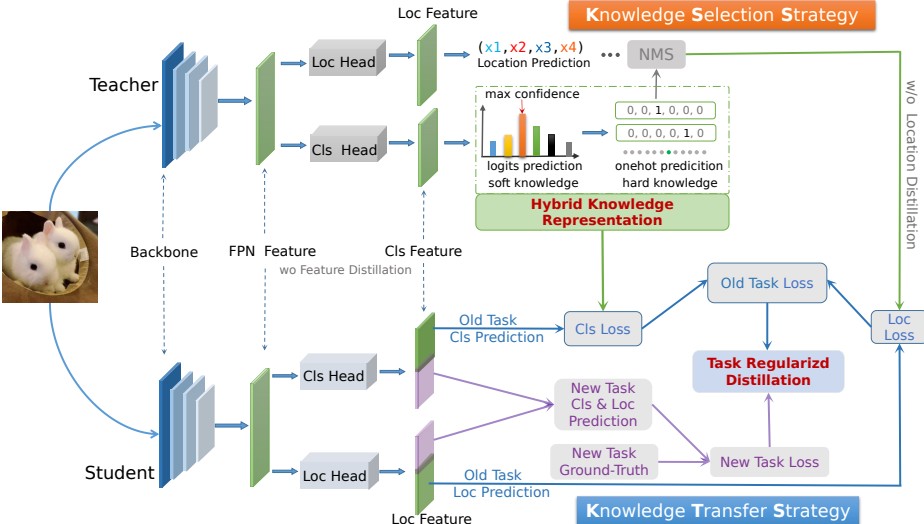

Figure 1: The overall architecture of our continual detector ilYOLOX. Cls and Loc refer to classification and location respectively. Hybrid Knowledge Representation (HKR, Eq.3) and Task Regularized Distillation (TRD, Eq.10) refer to our proposed two components respectively. HKR and TRD play roles of knowledge selection and knowledge transfer strategies respectively. During learning of student on new task, teacher is frozen and outputs predictions of old task. ILYOLOX achieves SOTA performance using only classification distillation without any feature and location distillation.

## 3.2 HYBRID KNOWLEDGE REPRESENTATION

Teacher outputs probability distribution as logits and converts them to one-hot labels as final predictions. Logits are regarded as soft knowledge, while one-hot labels are regarded as hard knowledge. Hinton et al. (2015) shows that soft knowledge is better than hard knowledge for classification distillation. However, although soft knowledge reflects more between-class information than hard knowledge, it also brings fuzziness to knowledge inevitably, which makes student confused during distillation learning. Meanwhile, teacher confidence reflects knowledge quality. If teacher has high confidence about its predictions, we should further strengthen this trend so that student can feel the certainty of this knowledge. Conversely, if teacher has low confidence, we should not do that.

Therefore, the key problem is how to evaluate the quality of soft knowledge from teacher. We here propose to evaluate soft knowledge according to the confidence difference between the maximum value and the secondary maximum value of teacher logits. Given a batch of images, teacher outputs a batch of logits for potential objects. For every logits in the batch, if the difference between the maximum confidence and the secondary maximum confidence is larger than a threshold, the knowledge quality of this logits will be regarded as high, otherwise as vanilla. High quality knowledge will be represented as one-hot prediction, while vanilla knowledge will be represented as soft prediction. We compute the mean value of the confidence differences across the entire batch as the threshold to judge knowledge quality adaptively. We formulate the description above as follows:

$$ConfDiff = Conf_{max} - Conf_{secondary\_max} \tag{1}$$

$$quality = ConfDiff > \frac{1}{N}\sum_{i}^{N} ConfDiff_i \tag{2}$$

$$Hybrid = quality \cdot Onehot + (1 - quality) \cdot Soft \tag{3}$$

where, $Conf^{N \times C}$ refers to a batch of logits predictions with batch size of $N$ and categories of $C$. $ConfDiff^{N \times 1}$ refers to the confidence difference for every logits between its maximum confidence and secondary maximum confidence. $N$ and $i$ refers to the total number of logits and the $i^{th}$ logit. $\frac{1}{N}\sum_{i}^{N} ConfDiff_i$ is the threshold to judge knowledge quality. $quality$ defined in Eq.2 is a Boolean vector to indicate knowledge quality. Then, $Hybrid$ predictions can be computed in Eq.3 by com-

bining $Onehot$ predictions and $Soft$ predictions. Obviously, our method combines soft knowledge and hard knowledge dynamically to form a hybrid knowledge representation for every input image.

### 3.3 TASK REGULARIZED DISTILLATION

The learning loss of student in COD task can be defined as following equation Eq.6. New task loss ($Loss_{new}$, Eq.5) refers to the loss supervised by the ground-truth of new classes. Old task loss ($Loss_{old}$, Eq.4) refers to the loss supervised by one-hot or soft targets from teacher. The $Loss_{cls}$ and $Loss_{loc}$ are the same as the official YOLOX, which are cross entropy loss and IoU loss respectively with coefficients of $\alpha = 1$ and $\beta = 5$. The task balance factor $\gamma$ is set to be 1 by default.

$$Loss_{old} = \alpha \cdot Loss_{cls} + \beta \cdot Loss_{loc} \tag{4}$$

$$Loss_{new} = \alpha \cdot Loss_{cls} + \beta \cdot Loss_{loc} \tag{5}$$

$$Loss_{total} = Loss_{new} + \gamma \cdot Loss_{old} \tag{6}$$

Continual learning is easily affected by data proportion of old and new tasks. If the data proportion of new task is too large, student will be dominated by new task loss and forget old knowledge. Conversely, student will obtain much more stability to old knowledge and lack of plasticity to accept new knowledge. Therefore, the key problem of distillation learning is to keep balance between old and new tasks. Motivated by this insight, we propose a **t**ask **r**egularized **d**istillation method (TRD) to solve the imbalance learning problem. TRD method consists of two parts: task equal loss and task difference loss, which are formulated as follows:

$$Loss_{old}^* = \frac{2 \cdot Loss_{new}}{Loss_{old} + Loss_{new}} \cdot Loss_{old} \tag{7}$$

$$Loss_{new}^* = \frac{2 \cdot Loss_{old}}{Loss_{old} + Loss_{new}} \cdot Loss_{new} \tag{8}$$

$$Loss_{diff}^* = (\frac{N_{new}}{N_{old}})^2 \cdot (Loss_{old} - Loss_{new})^2 \tag{9}$$

$$Loss_{total}^* = Loss_{new}^* + Loss_{old}^* + Loss_{diff}^* \tag{10}$$

Where, $Loss_{old}$ and $Loss_{new}$ are defined in Eq.4 and Eq.5, $N_{old}$ and $N_{new}$ refer to the number of old and new classes, respectively. $Loss_{old}^*$ and $Loss_{new}^*$ are the newly defined losses for old and new tasks, in which two dimensionless coefficients play a role of cross balancing factor. **Obviously, $Loss_{old}^*$ and $Loss_{new}^*$ will be always equal to each other during the entire continual learning, which ensures a completely dynamic balance between old and new tasks regardless of their data imbalance**. $Loss_{diff}^*$ measures the loss difference and categories proportion between old and new tasks, which can further contribute to their balance learning. $Loss_{total}^*$ is the final formulation of TRD method. Compared with Eq.6, TRD emphasizes task balance explicitly by introducing a task-balancing penalty item (Eq.9) and prevents student from over-fitting to any task. Fig.5 clearly compares their loss curves.

## 4 EXPERIMENTS

### 4.1 IMPLEMENTATION DETAILS

We build our continual detector on the top of YOLOX (Ge et al., 2021). It is a typical one-stage anchor-free detector among famous and widely used YOLO series, which can contribute to the typical verification of our method. YOLOX uses CSPNet (Wang et al., 2020) as its backbone, which needs to be trained from scratch along with detection heads for 300 epochs [1]. In general COD settings (Li et al., 2019), 1x training schedule with 12 epochs and frozen backbone are widely used. So we replace CSPNet with ResNet backbone (He et al., 2016) for 12 epochs training, which is pre-trained on ImageNet and frozen during continual learning. Official YOLOX adopts very strong data augmentation, including Mosaic, MixUP, Photo Metric Distortion, EMA, Random Affine, etc., to boost its performance, but we drop these tricks to reduce randomness. We keep the other components and hyper parameters of YOLOX unchanged. The modified YOLOX, denoted as **ilYOLOX**, is used for continual object detection. The ilYOLOX trained on old task is used as teacher to guide the next step learning of student on new task.

---

[1] seen YOLOX in https://github.com/open-mmlab/mmdetection

## 4.2 DATASETS AND EVALUATION METRICS

MS COCO2017 (Chen et al., 2015) is used to build benchmarks with the train set for training and the minival set for testing. We split the dataset of total 80 categories into several subsets by its alphabetic order. Each subset is a continual learning task. For example, the scenario of 40+20+20 means the data is split into three subsets and each of them contains 40, 20 and 20 categories respectively. The standard COCO protocols, including $AP$, $AP_{50}$, $AP_{75}$, $AP_S$, $AP_M$ and $AP_L$, are used to evaluate the detection performance.

In order to better evaluate the performance of COD task, we use following metrics. (1) **AbsGap** and **RelGap**: the absolute gap and relative gap between the task-average mAP of continual learning and the mAP of overall learning, respectively. The mAP of overall learning from baseline detector is usually considered as **Upper Bound mAP** (denoted as Upper or Upper Bound) of continual learning. RelGap equals to the ratio of AbsGap to Upper Bound mAP. (2) $\mathbf{F_b^1}$ and $\mathbf{F_b^2}$ , defined in Eq.11(a) and Eq.11(b) respectively, are proposed to evaluate the balance ability between old and new tasks. (3) **Omega ($\Omega$)**, defined in Eq.12 (Hayes et al., 2018), can evaluate the cumulative capability of continual learning task by task. Similar to COCO protocols, $\Omega$ can be extended as $\Omega_{all}$, $\Omega_{50}$, $\Omega_{75}$, $\Omega_S$, $\Omega_M$ and $\Omega_L$. **RelGap and $\Omega$ can eliminate the influence of upper bound from baseline detectors, providing a fair comparison among different continual learning methods**.

$$\mathbf{(a)}F_b^1 = \frac{|mAP_{old} - mAP_{new}|}{mAP_{old} + mAP_{new}} \qquad \mathbf{(b)}F_b^2 = \frac{2\sqrt{mAP_{old} \times mAP_{new}}}{mAP_{old} + mAP_{new}} \qquad (11)$$

$$\Omega = \frac{1}{T}\sum_{t=1}^{T}\frac{mAP_{continual,t}}{mAP_{overall,t}} \qquad (12)$$

where $T$ and $t$ is the number of total tasks and the $t^{th}$ task in continual learning. $mAP_{continual,t}$ and $mAP_{overall,t}$ means the task-average mAP and upper-bound mAP on all testing data containing learned categories after task $t$, respectively. The larger the $\Omega$ metric is, the better the ability of reducing cumulative knowledge forgetting would be.

## 4.3 EXPERIMENT SETUP

We design the following scenarios as benchmarks. **(i)Two-Task** scenarios: 40+40, 50+30, 60+20 and 70+10 settings. **(ii)Three-Task** scenario: 40+20+20 setting with 20 new classes added to the previous learned classes at each learning step. **(iii)Four-Task** scenario: 20+20+20+20 setting with 20 new classes added to the previous learned classes at each learning step. **(iiii)Five-Task** scenario: 40+10+10+10+10 setting with 10 new classes added to the previous learned classes at each learning step. We denote A(a-b) as the first-step overall learning for classes a-b, while +B(c-d) as the successive continual learning steps for classes c-d.

Given a scenario, we continually train ilYOLOX task by task (step by step) under the following settings. Optimizer is SGD with warm-up iterations of 1500, a continual learning rate of 0.2 decayed by 10% at the $8^{th}$ and $11^{th}$ epochs respectively, a momentum of 0.9 and a weight decay of 0.0005. All experiments are performed on 8 NVIDIA 3090 GPUs with a total batch size of 16×8. All images are randomly resized to [640 × 320, 640 × 640] by their short sides with content shape ratios unchanged. Normalization and random horizontal flip with a probability of 50% are also used.

## 4.4 OVERALL PERFORMANCE

### 4.4.1 CONTINUAL LEARNING ABILITY

Table.1, Table.2 and Table.3 report the results of Two-Task, Three-Task and Five-Task respectively. Compared with previous works, including LwF (Li & Hoiem, 2018), RILOD (Li et al., 2019), SID (Peng et al., 2021) and the latest best method ERD (Feng et al., 2022), our method achieves best performance under all scenarios and all evaluation metrics of continual learning. For the most difficult scenario of Five-Task (40+10+10+10+10) in Table.2, our method shows overwhelming advantages over ERD under final mAP (27.23% vs 20.70%), AbsGAP (7.03% vs 19.50%), RelGAP (20.53% vs 48.51%, 27.98 absolute improvement) and $\Omega_{all}$ (0.8893 vs 0.7961). Especially, though ERD has a higher initial mAP for A(1-40) and a higher upper bound mAP, but the final mAP (marked by

underline in Table.3 and Table.2) are reversed during continual learning. The results in Table.3 also show the same trend. We further plot $\Omega_{all}$ curves in Fig.2 to highlight our advantages. These results fully demonstrate much better continual learning capacity of our methods. In order to analysis the influence of upper bound mAP, we make extra experiments by using YOLOX-medium. Experiments in Table.9 (seen in AppendixA) demonstrate that our method get much better performance with a very large improvement under higher upper bound mAP. This further demonstrates its potential.

Table 1: Continual learning results under Two-Task scenarios. Our experiments are implemented with YOLOX-small, while all the others are implemented with GFLv1.

| Scenarios | Method | AbsGap↓ | RelGap↓ | $\Omega_{all}$ ↑ | $\Omega_{50}$ ↑ | $\Omega_{75}$ ↑ | $\Omega_S$ ↑ | $\Omega_M$ ↑ | $\Omega_L$ ↑ |
|---|---|---|---|---|---|---|---|---|---|
| | LwF | 23.00 | 57.21% | 0.7139 | 0.7178 | 0.7133 | 0.6703 | 0.7086 | 0.7328 |
| | RILOD | 10.30 | 25.62% | 0.8719 | 0.8859 | 0.8670 | 0.8405 | 0.8741 | 0.8879 |
| 40+40 | SID | 6.20 | 15.42% | 0.9229 | 0.9408 | 0.9163 | 0.8966 | 0.9354 | 0.9301 |
| | ERD | 3.30 | 8.21% | 0.9590 | 0.9674 | 0.9541 | 0.9591 | 0.9580 | 0.9550 |
| | Ours | **1.86** | **5.42%** | **0.9729** | **0.9688** | **0.9728** | **0.9659** | **0.9766** | **0.9697** |
| | LwF | 35.20 | 87.56% | 0.5622 | 0.5815 | 0.5528 | 0.6078 | 0.5760 | 0.5546 |
| | RILOD | 11.70 | 29.10% | 0.8545 | 0.8705 | 0.8463 | 0.8319 | 0.8583 | 0.8640 |
| 50+30 | SID | 6.40 | 15.92% | 0.9204 | 0.9374 | 0.9140 | 0.8793 | 0.9320 | 0.9320 |
| | ERD | 3.60 | 8.96% | 0.9552 | 0.9631 | 0.9461 | 0.9181 | 0.9580 | 0.9598 |
| | Ours | **2.03** | **5.92%** | **0.9704** | **0.9642** | **0.9741** | **0.9886** | **0.9805** | **0.9674** |
| | LwF | 34.40 | 85.57% | 0.5721 | 0.5926 | 0.5608 | 0.5862 | 0.5964 | 0.5738 |
| | RILOD | 14.80 | 36.82% | 0.8159 | 0.8328 | 0.8073 | 0.7996 | 0.8288 | 0.8228 |
| 60+20 | SID | 7.50 | 18.66% | 0.9067 | 0.9271 | 0.8968 | 0.8707 | 0.9263 | 0.9167 |
| | ERD | 4.40 | 10.95% | 0.9453 | 0.9537 | 0.9404 | 0.9440 | 0.9467 | 0.9454 |
| | Ours | **2.49** | **7.25%** | **0.9637** | **0.9578** | **0.9687** | **0.9801** | **0.9688** | **0.9640** |
| | LwF | 33.10 | 82.34% | 0.5883 | 0.6063 | 0.5803 | 0.6034 | 0.6077 | 0.5958 |
| | RILOD | 15.70 | 39.05% | 0.8047 | 0.8250 | 0.7947 | 0.8060 | 0.8107 | 0.8209 |
| 70+10 | SID | 7.40 | 18.41% | 0.9080 | 0.9202 | 0.9014 | 0.8685 | 0.9184 | 0.9262 |
| | ERD | 5.30 | 13.18% | 0.9341 | 0.9451 | 0.9289 | 0.9030 | 0.9399 | 0.9358 |
| | Ours | **3.14** | **9.16%** | **0.9542** | **0.9532** | **0.9564** | **0.9517** | **0.9596** | **0.9539** |

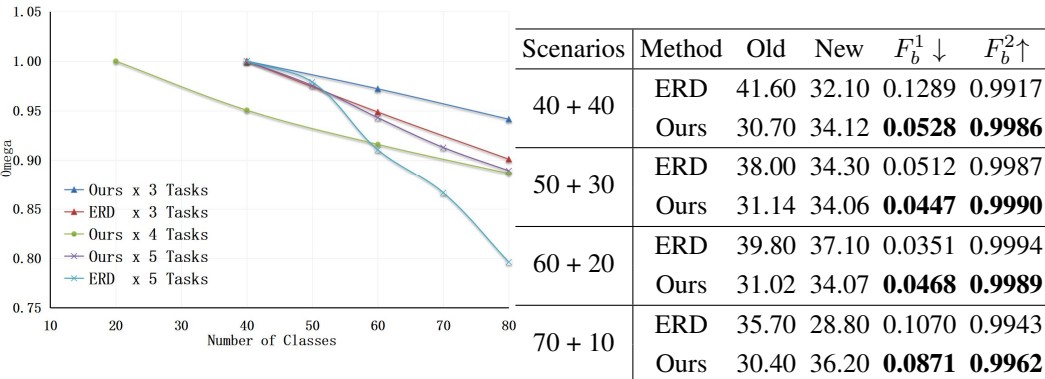

| Scenarios | Method | Old | New | $F_b^1$ ↓ | $F_b^2$ ↑ |
|---|---|---|---|---|---|
| 40 + 40 | ERD | 41.60 | 32.10 | 0.1289 | 0.9917 |
| | Ours | 30.70 | 34.12 | **0.0528** | **0.9986** |
| 50 + 30 | ERD | 38.00 | 34.30 | 0.0512 | 0.9987 |
| | Ours | 31.14 | 34.06 | **0.0447** | **0.9990** |
| 60 + 20 | ERD | 39.80 | 37.10 | 0.0351 | 0.9994 |
| | Ours | 31.02 | 34.07 | **0.0468** | **0.9989** |
| 70 + 10 | ERD | 35.70 | 28.80 | 0.1070 | 0.9943 |
| | Ours | 30.40 | 36.20 | **0.0871** | **0.9962** |

Figure 2: The performance of multi-task continual learning on MS COCO2017

Figure 3: The balance performance between old and new tasks on MS COCO2017.

### 4.4.2 BALANCE LEARNING ABILITY

For continual learning, the balancing ability between stability of old knowledge and plasticity of new knowledge is very important. Table.3 compares the results of ERD method and our method under the metrics of $\mathbf{F_b^1}$ and $\mathbf{F_b^2}$. Obviously, our method strikes a better balance between the mAP of old and new tasks, reflecting its good balancing ability between stability of old knowledge and plasticity

Table 2: Continual learning results under Five-Task scenario of 40+10+10+10. A(a-b) is the first-step normal training for categories a-b and +B(c-d) is the successive continual training for categories c-d. We use A(1-40) as the first step.

| | A(1-40) | | | | | | | |
|---|---|---|---|---|---|---|---|---|
| | +B(40-50) | | | | +B(50-60) | | | |
| | mAP | AbsGap↓ | RelGap↓ | $\Omega_{all}$ ↑ | mAP | AbsGap↓ | RelGap↓ | $\Omega_{all}$ ↑ |
| CF | 5.80 | 32.20 | 84.74% | 0.5763 | 5.70 | 34.10 | 85.68% | 0.4319 |
| RILOD | 25.40 | 12.60 | 33.16% | 0.8342 | 11.20 | 28.60 | 71.86% | 0.6499 |
| SID | 34.60 | 3.40 | 8.95% | 0.9553 | 24.10 | 15.70 | 39.45% | 0.8387 |
| ERD | 36.40 | **1.60** | **4.21%** | **0.9789** | 30.80 | 9.00 | 22.61% | 0.9106 |
| Ours | 32.47 | 1.69 | 4.94% | 0.9753 | 29.56 | **4.10** | **12.17%** | **0.9430** |
| | +B(60-70) | | | | +B(70-80) | | | |
| | mAP | AbsGap↓ | RelGap↓ | $\Omega_{all}$ ↑ | mAP | AbsGap↓ | RelGap↓ | $\Omega_{all}$ ↑ |
| CF | 6.30 | 29.40 | 82.35% | 0.3681 | 3.30 | 36.90 | 91.79% | 0.3109 |
| RILOD | 10.50 | 25.20 | 70.59% | 0.5610 | 8.40 | 31.80 | 79.10% | 0.4906 |
| SID | 14.60 | 21.10 | 59.10% | 0.7313 | 12.60 | 27.60 | 68.66% | 0.6477 |
| ERD | 26.20 | 9.50 | 26.61% | 0.8664 | 20.70 | 19.50 | 48.51% | 0.7961 |
| Ours | 27.59 | **5.94** | **17.72%** | **0.9129** | 27.23 | **7.03** | **20.53%** | **0.8893** |

of new knowledge. The ablation experiments and supplementary material show more analyses about task-balancing learning.

### 4.4.3 KNOWLEDGE FORGETTING

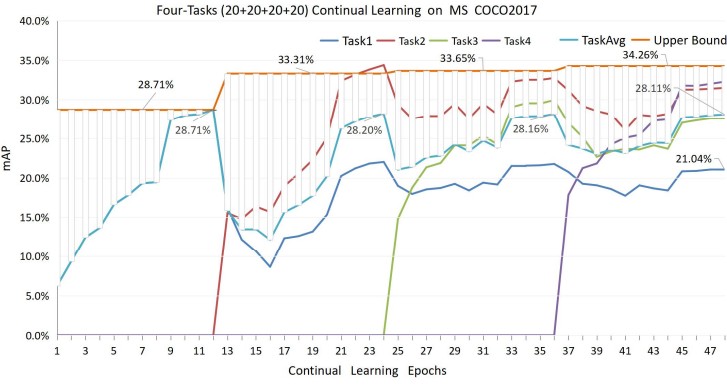

Figure 4: The continual learning curves under Four-Task scenario (20+20+20+20), which shows that knowledge forgetting of old tasks is always controlled to a limited range during learning process.

Fig.4 reports the results under Four-Task scenario, which clearly shows detailed continual learning process of ilYOLOX in successive data streams. The AbsGap (plotted as white fluctuation columns at each learning epoch) shows that our method reduces the performance gap between continual learning and overall learning gradually and effectively. The mAP curves of each task (denoted as $Task_i$, $i = 1, 2, 3, 4$) reflect their knowledge forgetting process. On the whole, the knowledge forgetting of old tasks is always controlled to a limited range during successive continual learning. Meanwhile, the TaskAvg mAP after each learning step, marked on the corresponding light-blue curve, also shows very good stability (28.71%, 28.20%, 28.16%, 28.11%). The results in Fig.4 further demonstrate good effectiveness of our method.

## 5 ABLATION STUDY

**The Independence and Compatibility of HKR and TRD.** Table.4 shows the results of ablation experiments. The two baseline methods, denoted as Onehot and Soft, use one-hot predictions and soft

Table 3: Continual learning results under Three-Task scenario of 40+20+20.

| | A(1-40) | | | | | | | |
|---|---|---|---|---|---|---|---|---|
| | +B(40-60) | | | | +B(60-80) | | | |
| | mAP | AbsGap | RelGap↓ | $\Omega_{all}$ ↑ | mAP | AbsGap | RelGap↓ | $\Omega_{all}$ ↑ |
| CF | 10.70 | 29.10 | 73.38% | 0.6344 | 9.40 | 30.80 | 76.62% | 0.5009 |
| RILOD | 27.80 | 12.00 | 30.85% | 0.8492 | 15.80 | 24.40 | 60.70% | 0.6972 |
| SID | 34.00 | 5.80 | 15.42% | 0.9271 | 23.80 | 16.40 | 40.80% | 0.8154 |
| ERD | 36.70 | 3.10 | 7.79% | 0.9611 | 32.40 | 7.80 | 19.40% | 0.9094 |
| Ours | 32.26 | **1.39** | **4.13%** | **0.9793** | 31.24 | **3.02** | **8.82%** | **0.9568** |

Table 4: Continual learning results under Two-Task scenarios for ablation study. We equip YOLOX-small with two knowledge selection methods (Onehot and Soft) as our baselines.

| Scenarios | 60 classes + 20 classes | | | | | 70 classes + 10 classes | | | | |
|---|---|---|---|---|---|---|---|---|---|---|
| Methods | $AbsGap \downarrow$ | $RelGap \downarrow$ | $\Omega_{all}$ ↑ | $\Omega_{50}$ ↑ | $\Omega_{75}$ ↑ | $AbsGap \downarrow$ | $RelGap \downarrow$ | $\Omega_{all}$ ↑ | $\Omega_{50}$ ↑ | $\Omega_{75}$ ↑ |
| Onehot | 3.86 | 11.27% | 0.9437 | 0.9477 | 0.9387 | 4.91 | 14.33% | 0.9284 | 0.9367 | 0.9264 |
| Soft | 3.29 | 9.60% | 0.9520 | 0.9505 | 0.9537 | 4.49 | 13.10% | 0.9345 | 0.9385 | 0.9332 |
| Soft+HKR | 3.10 | 9.04% | 0.9548 | 0.9514 | 0.9578 | 4.19 | 12.22% | 0.9389 | 0.9422 | 0.9414 |
| Soft+TRD | 2.95 | 8.62% | 0.9569 | 0.9541 | 0.9578 | 4.06 | 11.83% | 0.9408 | 0.9431 | 0.9441 |
| Soft+Both | **2.49** | **7.26%** | **0.9637** | **0.9578** | **0.9687** | **3.14** | **9.16%** | **0.9542** | **0.9532** | **0.9564** |

predictions (logits) as teacher knowledge respectively. Then we add Hybrid Knowledge Selection module and Task Regularized Distillation module to the Soft baseline respectively (seen in Fig.1), whose results are denoted as Soft+HKR and Soft+TRD respectively. Finally, we add both HKR and TRD to the Soft baseline simultaneously, whose results are denoted as Soft+Both. The results under two scenarios all show that soft knowledge is better than hard knowledge, but both are inferior to hybrid knowledge. Compared with the Soft baseline, TRD shows higher performance improvement than HKR. This demonstrates that both HKR and TRD have their own effects as two independent components. Meanwhile, the results of 'Soft+Both' (means Soft+HKR+TRD) get further significant improvement, demonstrating that HKR and TRD have good additivity and compatibility.

**The HKR under Three-Task Scenario.** In order to further analyze hybrid knowledge representation, we make additional ablation studies under Three-Task scenario. The results shown in Table.5 demonstrate the effectiveness of HKR clearly. It further reflects that hybrid knowledge can fully utilize the advantages of both soft knowledge and hard knowledge in adaptive manner.

**The Task Balance During Continual Learning.** In order to further analyze the influence of task balance on continual learning, we make experiments by changing the task balancing factor ($\gamma$ in Eq.6) from 0.2 to 3.0. The results of Table.6 show that our TRD method gets a medium mAP for new task, but gets the highest performance under all other metrics. Brown-marked digits show the sub-optimal results. When $\gamma$ changes from 0.2 to 3.0, the mAP values in Table.6 shows noticeable changes. A similar experiment is made on another small dataset (seen appendix) and shown in Fig.6. It shows that even a very small change of $\gamma$ from 1 to 0.9 can lead to dramatically descending of the old task mAP curve (blue dotted line) and bring catastrophic knowledge forgetting. Obviously, task balance factor ($\gamma$) has a significant influence on continual learning by controlling knowledge

Table 5: Continual learning results under Three-Task scenario (40+20+20) for ablation study.

| Scenarios | +B(40-60) | | | | | | +B(60-80) | | | | | |
|---|---|---|---|---|---|---|---|---|---|---|---|---|
| methods | mAP | AbsGap | RelGap↓ | $\Omega_{all}$ ↑ | $\Omega_{50}$ ↑ | $\Omega_{75}$ ↑ | mAP | AbsGap | RelGap↓ | $\Omega_{all}$ ↑ | $\Omega_{50}$ ↑ | $\Omega_{75}$ ↑ |
| $Onehot$ | 31.62 | 2.04 | 6.05% | 0.9698 | **0.9721** | 0.9694 | 30.44 | 3.83 | 11.16% | 0.9426 | 0.9453 | 0.9424 |
| $Soft$ | 31.84 | 1.81 | 5.38% | 0.9731 | 0.9712 | 0.9750 | 30.87 | 3.39 | 9.91% | 0.9491 | 0.9453 | 0.9488 |
| $Hybrid$ | **32.03** | **1.62** | **4.81%** | **0.9759** | 0.9703 | **0.9819** | **31.19** | **3.08** | **8.97%** | **0.9540** | **0.9478** | **0.9580** |

transfer from teacher to student. Compared with different $\gamma$ values, the mAP of old 70 classes reflects that TRD relieves knowledge forgetting to the greatest extent. Meanwhile, TRD shows much more balance learning ability between old and new tasks with the minimum $F_b^1$ and the maxmum $F_b^2$. **Overall speaking, our TRD realizes dynamic balance knowledge transfer by introducing task-based regularization. More over, TRD is a statistically adaptive method without hyper parameters**.

Table 6: Continual learning results under Two-Task scenario for ablation study. Hyper parameter $\gamma$ is the task balance factor (seen in Eq.6). TRD is task regularized distillation method (seen in Eq.10)

| Methods | mAP ↑ | | | $\Omega_{all} \uparrow$ | $\Omega_{50} \uparrow$ | $\Omega_{75} \uparrow$ | $F_b^1 \downarrow$ | $F_b^2 \uparrow$ |
|---|---|---|---|---|---|---|---|---|
| | Old 70 Classes | New 10 Classes | Final | | | | | |
| $\gamma = 0.2$ | 27.00 | 37.41 | 28.30 | 0.9129 | 0.9257 | 0.9101 | 0.1617 | 0.9868 |
| $\gamma = 0.5$ | 28.41 | 37.70 | 29.57 | 0.9315 | 0.9367 | 0.9305 | 0.1406 | 0.9901 |
| $\gamma = 0.6$ | 28.37 | **38.02** | 29.58 | 0.9316 | 0.9358 | 0.9292 | 0.1453 | 0.9894 |
| $\gamma = 0.7$ | 28.58 | 37.69 | 29.72 | 0.9336 | 0.9385 | 0.9319 | 0.1375 | 0.9905 |
| $\gamma = 0.8$ | 28.92 | 37.48 | 29.99 | 0.9376 | 0.9394 | 0.9360 | 0.1289 | 0.9917 |
| $\gamma = 0.9$ | 28.59 | 37.53 | 29.71 | 0.9335 | 0.9358 | 0.9332 | 0.1352 | 0.9908 |
| $\gamma = 1.0$ | 28.72 | 37.14 | 29.78 | 0.9345 | 0.9385 | 0.9332 | 0.1278 | 0.9918 |
| $\gamma = 1.5$ | 28.61 | 36.96 | 29.66 | 0.9328 | 0.9358 | 0.9292 | 0.1273 | 0.9919 |
| $\gamma = 2.0$ | 28.37 | 36.04 | 29.33 | 0.9280 | 0.9312 | 0.9292 | 0.1191 | 0.9929 |
| $\gamma = 3.0$ | 27.11 | 35.14 | 28.12 | 0.9103 | 0.9174 | 0.9074 | 0.1289 | 0.9917 |
| TRD | **29.24** | 36.97 | **30.21** | **0.9408** | **0.9431** | **0.9441** | **0.1167** | **0.9932** |

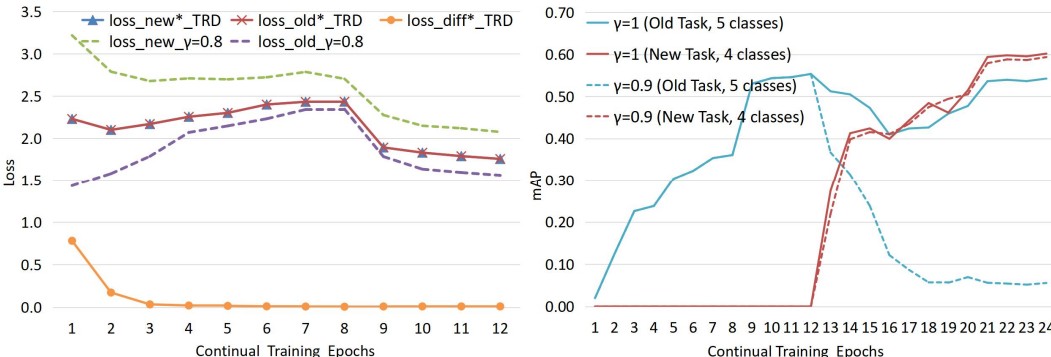

Figure 5: The training loss of old and new tasks at $\gamma = 0.8$ and TRD for Table.6 respectively.

Figure 6: The catastrophic forgetting caused by imbalance learning of old and new tasks.

## 6  CONCLUSION

In order to improve the performance of continual object detection, we propose a knowledge distillation method that combines knowledge selection strategy and knowledge transfer strategy effectively. For the first strategy, hard knowledge and soft knowledge are dynamically and adaptively combined to construct a kind of hybrid knowledge representation to use teacher knowledge critically and effectively. For the second strategy, loss difference and category proportion are combined to construct task regularized distillation loss to enhance task balance learning. Extensive experiments under different scenarios validate the effectiveness of our method. Most existing methods in COD task adopt a mixed distillation scheme including feature, classification, location and relation to relieve catastrophic forgetting. However, we demonstrate that as long as knowledge selection and transfer strategies are appropriate, even single classification distillation can also achieve state-of-the-art performance. In addition, our method has a good prospect to work together with feature and location distillation for further improvements. More analyses and discussions are provided in Appendix A.

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

## A APPENDIX

### A.1 COMPARED WITH KL DIVERGENCE LOSS

Kullback-Leibler Divergence loss (denoted as KLD loss) is used for knowledge distillation of image classification (Hinton et al., 2015). YOLOX (Ge et al., 2021) uses cross entropy loss (denoted as CE loss) for its classification head. Table.7 shows the comparison results of this two losses on continual object detection. The experiments adopt the same loss weight setting with $\alpha = 1$ and $\beta = 5$ (seen in Eq.4 and Eq.5) for the two losses. $T$ is temperature factor, a hyper parameter of KLD loss. When the temperature $T$ changes from 1 to 5, ilYOLOX gets its best performance (marked by brown color) under the medium temperature of 3. However, CE loss has a much better performance than KLD loss (marked by underline). Our TRD loss exceeds both KLD and CE loss. The use of KLD loss usually requires careful adjustment of temperature factor $T$ and loss weight $\alpha$. However, the change of $\alpha$ in $Loss_{old}$ will destroy the loss consistency about classification and location between old task ($Loss_{old}$, Eq.4) and new task ($Loss_{new}$, Eq.5), which will influence task balance during continual learning. Based on this consideration and the experiment results in Table.7, we use cross entropy loss as our fundamental knowledge transfer strategy.

Table 7: The comparison with KL Divergence Loss

| Scenarios | 70 classes + 10 classes | | | | | |
|-----------|-------------|---------------|-------|-----------|-----------|------------------|
| Methods | $mAP \uparrow$ | | | $AbsGap \downarrow$ | $RelGap \downarrow$ | $\Omega_{all} \uparrow$ |
| | Old 70 Classes | New 10 Classes | Final | | | |
| KLD T=1 | 25.55 | 35.19 | 26.75 | 7.51 | 21.92% | 0.8904 |
| KLD T=2 | 26.36 | 35.87 | 27.55 | 6.71 | 19.59% | 0.9021 |
| KLD T=3 | 26.86 | 36.29 | 28.04 | 6.22 | 18.16% | 0.9092 |
| KLD T=4 | 26.84 | 35.96 | 27.98 | 6.29 | 18.35% | 0.9083 |
| KLD T=5 | 26.49 | 35.50 | 27.62 | 6.65 | 19.41% | 0.9030 |
| CE Loss | 28.72 | **37.14** | 29.78 | 4.49 | 13.10% | 0.9345 |
| TRD Loss | **29.24** | 36.97 | **30.02** | **4.24** | **12.38%** | **0.9381** |

## A.2 Experiment Details of Fig.6

We describe the experiments in Fig.6 in detail as follows. It is made on a small dataset that consists of 3800 images, 9 classes (commonly seen toys including car, truck, train, person and so on) and have $400 \sim 500$ instances for every class with a relative balanced category distribution. We build a Two-Task scenario of 5 classes + 4 classes as our continual object detection benchmark. The experiment results are illustrated in Table.8. When the task balance factor ($\gamma$ defined in Eq.6) changes from 1.3 to 0.7, the mAP of old task (old 5 classes) drops from 54.24% to 5.10% quickly. It fully demonstrates that the loss imbalance between old and new tasks can bring significant catastrophic forgetting during continual leaning.

Table 8: Task balance experiment on a small dataset

| Scenarios | 5 classes + 4 classes | | | | | |
|-----------|-------------|---------------|-------|-----------|-----------|------------------|
| Methods | mAP$\uparrow$ | | | AbsGap$\downarrow$ | RelGap$\downarrow$ | $\Omega_{all} \uparrow$ |
| | Old 5 Classes | New 4 Classes | Final | | | |
| $\gamma = 0.7$ | 5.16 | 58.73 | 28.97 | 32.31 | 52.73% | 0.7364 |
| $\gamma = 0.9$ | 5.10 | 59.20 | 29.14 | 32.14 | 52.44% | 0.7378 |
| $\gamma = 1.0$ | 54.24 | 60.18 | 56.88 | 4.40 | 7.18% | 0.9641 |
| $\gamma = 1.1$ | 54.34 | 59.98 | 56.84 | 4.44 | 7.25% | 0.9638 |
| $\gamma = 1.3$ | 53.58 | 59.05 | 56.01 | 5.27 | 8.60% | 0.9570 |
| TRD | 54.46 | 64.18 | 58.78 | 2.50 | 4.08% | 0.9796 |
| TRD+HKR | **54.66** | **64.63** | **59.09** | **2.19** | **3.58%** | **0.9821** |

## A.3 Discussion

**Hybrid Knowledge Representation.** Teacher outputs logits and one-hot representation as its predictions. Soft logits contains more information about between-class confidences and is regarded as a kind of soft knowledge (Hinton et al., 2015; Yang et al., 2022c). Knowledge distillation methods are developed prosperously under this background in image classification and object detection tasks. But there are significant difference between the two tasks. For image classification, since an input image contains only one instance, the knowledge of the image is just the knowledge of its instance. However, for object detection, one input image contains several instances, the knowledge on an image is a collection of the knowledge of all instances on it. Instances on an image can be regarded as entity nodes, so the overall image knowledge should be constructed based on the entity nodes and their relations. In other words, image-level knowledge is the combination of instance-level knowledge. Logits and One-hot predictions provide two kinds of instance-level knowledge representation as soft and hard knowledge, respectively. Soft knowledge contains confidence relations among categories, but brings knowledge fuzziness inevitably. While, hard knowledge has completely opposite effects. By combining their advantages, we construct an image-level hybrid knowledge representation (Eq.3) showing better performance on COD task in Table.4 and Table.5.

**Task Regularized Distillation.** Regularization method is very import for statistical machine learning, which can prevent model from over-fitting to some part of data. Classical regularization methods introduce a weight constraint in terms of p-Norm as model penalties. Other regularization methods include stopping the training as soon as performance on a validation set starts to get worse and soft weight sharing (Nowlan & Hinton, 1992). Dropout works by dropping some connections randomly to prevent over-fitting of deep neural networks (Srivastava et al., 2014). In continual learning, model needs to be trained task by task in the continuous data flow, therefore, bring task-based imbalance learning and lead to catastrophic forgetting. A few previous works propose regularization-based methods on continual image classification (Kirkpatrick et al., 2017; Li & Hoiem, 2018). We give a solution by proposing task-based regularized distillation loss 10 for COD tasks. It explicitly uses the loss and categories difference between old and new tasks as a model penalty to constraint optimization process. Experiments in Table.4, Table.6, Table.8, Fig.5 and Fig.6 all show its strong effectiveness to prevent continual model from over-fitting to some task.

### A.4    More Details of Implementation and Experiments

All the previous works, including Catastrophic Forgetting, RILOD (Li et al., 2019), SID (Peng et al., 2021) and ERD (Feng et al., 2022) models, are implemented based on GFLv1 (Zheng et al., 2021) with original image size of $1333 \times 800$. Since they use the same base detector, the upper bound mAP of these previous models are all 40.20 for A(1-80), the overall training of total 80 categories. On the other hand, our experiments are implemented based on YOLOX-small detector (Ge et al., 2021). YOLOX is a typical one-stage anchor-free detector among famous and widely used YOLO series, which can contribute to the typical verification of our method. The official YOLOX implementation in MMDetection [2] adopts 300 epochs training schedule with strong data augmentation, including Mosaic, MixUP, Photo Metric Distortion, EMA, Random Affine, Random Horizontal Flip, etc, to boost its performance. We drop these tricks to reduce randomness for continual learning and for better reproducibility. The modified YOLOX, denoted as **ilYOLOX**, is used for continual object detection. Since our experiments are conducted for demonstration of model effectiveness, we randomly resize all the images to $[640 \times 320, 640 \times 640]$ by their short sides with content shape ratios unchanged. The upper bound mAP of our model is therefore 37.13 for A(1-80).

Although different models have different baseline detectors, the evaluation metrics, like $RelGap$ and $\Omega$, are used to fairly compare their continual learning capability. We also analyze the influence of upper bound mAP by comparing different base detectors: YOLOX-small versus YOLOX-medium. Experiments in Table.9 shows that higher upper bound mAP leads to better continual learning performance, which further demonstrates the potential of our method. Here, we bring the continual object detection research on widely used lightweight model YOLOX to reduce computing loads and resource consumption, accelerating the researches of this topic and protecting our environment.

Table 9: Results on COCO benchmark with different upper bound mAP.

| Scenarios | Method | AbsGap | RelGap↓ | $\Omega_{all}$ ↑ | $\Omega_{50}$ ↑ | $\Omega_{75}$ ↑ | $\Omega_S$ ↑ | $\Omega_M$ ↑ | $\Omega_L$ ↑ | mAP | Upper |
|---|---|---|---|---|---|---|---|---|---|---|---|
| 40 + 40 | ilYOLOX-S | 1.86 | 5.42% | 0.9729 | 0.9688 | 0.9728 | 0.9659 | 0.9766 | 0.9697 | 32.41 | 34.26 |
| | ilYOLOX-M | **1.11** | **2.99%** | **0.9849** | **0.9850** | **0.9887** | **0.9922** | **0.9915** | **0.9698** | 36.02 | 37.13 |
| 70 + 10 | ilYOLOX-S | 3.14 | 9.16% | 0.9542 | 0.9532 | 0.9564 | 0.9517 | 0.9596 | **0.9539** | 31.13 | 34.26 |
| | ilYOLOX-M | **2.62** | **7.04%** | **0.9648** | **0.9575** | **0.9699** | **0.9560** | **0.9732** | 0.9435 | 34.52 | 37.13 |

---

**Algorithm 1** Algorithm of Hybrid Knowledge Representation

---

**Input**: Unlabeled image $I$, teacher detector $\theta'$
**Output**: Hybrid predictions $Hybrid$ of response

 1: Inference $I$ with $\theta'$ yields the logits predictions $Soft$ and one-hot predictions $Onehot$
 2: Compute $ConfDiff = Conf_{max} - Conf_{secondary\_max}$
 3: Compute $quality = ConfDiff > \frac{1}{N}\sum_i^N ConfDiff_i$
 4: Compute $Hybrid = quality \cdot Onehot + (1 - quality) \cdot Soft$

---

[2] seen YOLOX in https://github.com/open-mmlab/mmdetection

