# OpenReview forum: "Task Regularized Hybrid Knowledge Distillation For Continual Object Detection"
_ICLR.cc/2023/Conference — Submitted to ICLR 2023_

### Official Review · Reviewer_jG8i · 2022-10-23

**Confidence:** 2
**Correctness:** 3
**Technical Novelty And Significance:** 2
**Empirical Novelty And Significance:** Not applicable
**Recommendation:** 3

**Clarity, Quality, Novelty And Reproducibility:**

Clarity: 6/10

Quality: 4/10

Novelty: 4/10

Reproducibility: Unknown

**Strength And Weaknesses:**

1. The fusion of soft and hard labels has been explored in [1], what is the difference between the proposed “HKS” and the fusion method of [1]? Can the authors additionally compare the fusion method of [1] in the experiment part?

2. Class-balanced loss is a common practice, such as the balanced CE loss in [2]. The idea behind the loss design of "TRD" seems not strong enough to be accepted.

3. Although the authors have repeatedly emphasized that different detection models do not affect some metrics like “RelGap” and “Omega”. I still think that the authors should consider experiments based on the same model as other methods.

4. I would like to know if the training settings of this paper is consistent with the settings of other papers used as comparisons. If consistent, some citations can be added to the part of experimental setup.

5. In Figure 4, the continual learning curves of other methods are expected to be plotted for better comparison.

6. The importance of “$Loss_{diff}$” term in “TRD” is not considered in ablations.

7. Can the observed phenomenon be reproduced when experiments w.r.t. Fig. 6 being carried in a common dataset, such as MS-COCO, rather than the designed small dataset?

Things to improve the paper that do not impact the score:

1. The loss formulation for “TRD” seems to be redundant, for example, only one of “$Loss_{new}^{\*}$” and “$Loss_{old}^{\*}$” is required in Eq.10.

2. Illustrations in the paper had better be vector graphs.

3. In the caption of Table 2, an additional “+10” is missing after “Five-Task scenario of 40+10+10+10”.

4. In Page 8, “Knowledge Selection” in “Then we add Hybrid Knowledge Selection module……” should be “Knowledge Representation” for consistency.

5. Some explanations in A.3, such as the instance-level discussions, seem to be unrelated to the main ideas of this paper and can be simply removed in my view.

[1]: Kim, Kyungyul, et al. "Self-knowledge distillation with progressive refinement of targets." Proceedings of the IEEE/CVF International Conference on Computer Vision. 2021.

[2]: Homayounfar, Namdar, et al. "VideoClick: Video Object Segmentation with a Single Click." arXiv preprint arXiv:2101.06545 (2021).




**Summary Of The Paper:**

This paper aims to alleviate the problem of catastrophic forgetting in continual object detection from the perspective of knowledge distillation (KD). Different from other methods that adopt complex KD supervision, such as feature, location and relation, the proposed method only relies on classification supervision and can also achieve promising results with the help of a combination of soft-hard label assignment and class-adaptive loss coefficient.

**Summary Of The Review:**

This paper should be rejected for the aforementioned reasons.

---

### Official Review · Reviewer_rDxx · 2022-10-24

**Confidence:** 4
**Correctness:** 2
**Technical Novelty And Significance:** 2
**Empirical Novelty And Significance:** 2
**Recommendation:** 3

**Clarity, Quality, Novelty And Reproducibility:**

The paper's clarity may be improved. In particular, the presentation of the results is hard to follow and it is not always evident which is the setting for the tables. Moreover, the methodological section should be improved following the suggestions in the previous answer.


**Strength And Weaknesses:**

Weaknesses:
1. The TRD is not clear from the paper's description. First, Eq.7 and Eq.8 are not only always equal, but without other explanations, they are exactly the same equation and have the exact same effect on the network gradients, actually being a single loss with value Loss*_old + Loss*_new = 4*Loss_new*Loss_old / (Loss_old + Loss_new). Seen in this perspective, TRD does not bring any contribution but just computes the harmonic mean between the two losses. Furthermore, very few intuitions are provided for Eq. 9. Why should minimizing the quadratic difference between the old and new loss be beneficial for the training? Finally, the paper should experimentally demonstrate the benefits of adding Loss_diff through an ablation study.
2. Mixing soft and hard pseudo-labeling is an interesting concept, however, to better exploit the model confidence, previous works propose to introduce a temperature value in the softmax operation to better transfer the model confidence. This is similar in spirit to the HKR and should be carefully demonstrated that the paper technique is superior. Furthermore, the paper introduces a threshold to decide whether use one-hot or soft-pseudo labeling. How is the threshold fixed? Is there any ablation study demonstrating the robustness of this choice?
3. Regarding the experimental section, the paper reports experiments on MS-COCO. However, the standard and mainly used benchmark for continual object detection is the Pascal-VOC dataset but it is not reported. The paper should report results on this benchmark and should compare with all the state-of-the-art methods and not only a subset of them.

**Summary Of The Paper:**

The paper proposes a knowledge distillation approach for continual object detection. The paper introduces two contributions: an image-level hybrid knowledge representation method, named HKR, that distill knowledge from the teacher by combining soft and hard pseudo-labels. Secondly, it proposes task regularized distilliation (TRD), that is a strategy to balance the knowledge distillation and classification loss, to prevent the student model to overfit the new tasks.
Overall, the method performance are assessed on MS-COCO using the YOLOX architecture.

**Summary Of The Review:**

The paper proposes a method for Continual Object Detection introducing two slight modifications to the standard knowledge distillation framework. Moreover, the paper lacks important ablation studies and comparisons with state-of-the-art results.

---

### Official Review · Reviewer_i3PC · 2022-10-30

**Confidence:** 5
**Correctness:** 2
**Technical Novelty And Significance:** 2
**Empirical Novelty And Significance:** 3
**Recommendation:** 3

**Clarity, Quality, Novelty And Reproducibility:**

Some suggestions which may make the paper clear:
1. Do not use the abbreviation when the term appears the first time, e.g. LWF.
2. Some terms are not well-aligned. e.g.: ilYOLOX and ILYOLOX
3. Figure 1 is confusing. If the FPN feature Is not used, there is no need to add the arrows between them. Has location distillation been employed? What is the meaning of 'w/o' and 'wo'.  If they all mean without, there is no need to include them in the feature.

**Strength And Weaknesses:**

++ The paper pays attention to the regularization of various tasks.
++The experiment results on multiple settings demonstrate the effectiveness of the proposed detectors.

-- The comparison happens between different detectors and training schemes, which is debatable. Although we should not be limited to old benchmarks for algorithm design, in order to make a fair comparison, experiments on the same detectors should first be shown. For example, the author could re-implement other continuous learning algorithms with the same detector, YOLOX. Or, the proposed method can be first applied to original benchmarks to make a fair comparison, and then a new powerful detector could be proposed for future comparison.

--Ablation studies can not fully demonstrate the effectiveness of the proposed TRD method. If there are more task stages in the training set, will the proposed TRD be more powerful? Or it performs the same. It will be better if more weight-balancing algorithms could be compared with TRD.

-- The representation is not that clear, including the terms and the figures. Details could be referred to in the next blank.

-- For the 60+20 setting in Table1 and Table 4. The one-hot performance of the YOLOX already outperforms the previous work ERD. The improvement of the proposed two methods is not that significant.



**Summary Of The Paper:**

The paper proposed a knowledge selection module between soft and hard labels. It also proposed a task-based regularization distillation loss to balance multiple task losses.

As a new detector, YOLOX, is employed in this paper, the performance of this work is higher than previous ones. Extensive results on multiple datasets have been shown.

**Summary Of The Review:**

I tend to reject the paper. The main contribution is to employ a new powerful detector YOLOX for continuous learning. This can not bring too many insights. I might improve my rating if the author could analysis some interesting insights related to their proposed methods.

---

### Official Review · Reviewer_qEVh · 2022-11-01

**Confidence:** 4
**Correctness:** 2
**Technical Novelty And Significance:** 2
**Empirical Novelty And Significance:** 2
**Recommendation:** 3

**Clarity, Quality, Novelty And Reproducibility:**

- Clarity and Quality: the paper has multiple places to be fixed and clarified (see my comments in weakness section).
- Novelty: the paper is technically marginal. Not an innovative idea, but rather an incremental improvement over some prior arts.
- Reproducibility: cannot tell.

**Strength And Weaknesses:**

**Strengths**
- S1: In under explored COD task, on COCO dataset, the gain by the proposed method is noticeable (Table 1)

**Weaknesses**
- W1: The hybrid equation (Eq. (1)-(3)) is not well motivated. Why $ConfDiff$ should be defined in this manner? Why should the hybrid loss mix up the onehot and soft in this manner (necessariliy)?
- W2: The eq (7) and (8) are the same. Explanation is required.
- W3: No justification of the proposed evaluation metric of $F_b^1$ and $F_b^2$ in Eq. (11).
- W4: Choice of hyperparameter also seems arbitrary or based on empirical results which may be specific to COCO dataset only. This limits the applicability of the proposed method to other datasets or tasks.

**Summary Of The Paper:**

The paper address the continual object detection problem, which is a very recent and challenging vision task. It proposes an image-level hybrid knowledge distillation method. The proposed method exhibits a noticeable gain in COCO 2017 dataset.

**Summary Of The Review:**

Given that the paper has only merit in empirical results but poorly motivates the proposed method, the paper should be heavily revised and encouraged to resubmit.

---

### Decision · Program_Chairs · 2023-01-20

**Decision:**

Reject

**Justification For Why Not Higher Score:**

The reviewers are unanimous in their opinion and specific points of concern (novelty, clarity, and experimental evaluation).

**Justification For Why Not Lower Score:**

N/A

**Metareview: Summary, Strengths And Weaknesses:**

# Summary of Contribution
This paper proposes an approach to continual learning of object detectors. The authors propose to use knowledge distillation method that uses hard and soft pseudo-labels, and a task-regularized distillation scheme that tries to balance the knowledge distillation and classification losses during continual learning. The authors report results on MS-COCO 2017 using their proposed YOLOX architecture.

# Strengths

Continual object detection is an important and challenging continual learning task, and the proposed approach along with the YOLOX architecture seems to perform well on MS-COCO.

# Weaknesses

+ **Clarity**: All reviewers pointed to significant problems with clarity in the technical presentation of the proposed method and the experimental results.
Novelty: the paper is technically marginal. Not an innovative idea, but rather an incremental improvement over some prior arts.

+ **Experimental Comparison**: Reviewers point to problems with ablation studies (that do not support the claims about the proposed approach made in the paper), missing standard datasets for continual object detection (Pascal VOC), and differences between the different detectors with different training protocols.

+ **Novelty**: The reviewers are unanimous in their opinion that the submitted manuscript represents a minor incremental improvement over prior works.

# Summary

Given the unanimous consensus of the reviewers, and the lack of author rebuttal to their concerns, the final conclusion is that the paper does not meet the novelty and quality bars for acceptance at ICLR. The authors are encouraged to use the reviewer comments to improve their manuscript, in particular to emphasize the novel aspects of the contribution and to align the experimental evaluation with standard practices in continual object detection.